# Measures of Entropy to Characterize Fatigue Damage in Metallic Materials

**DOI:** 10.3390/e21080804

**Published:** 2019-08-17

**Authors:** Huisung Yun, Mohammad Modarres

**Affiliations:** Center for Risk and Reliability, Department of Mechanical Engineering, University of Maryland, College Park, MD 20742, USA

**Keywords:** physics of failure, prognosis and health management, entropy as damage, fatigue, entropy generation, acoustic emission, information entropy, thermodynamic entropy, Jeffreys divergence

## Abstract

This paper presents the entropic damage indicators for metallic material fatigue processes obtained from three associated energy dissipation sources. Since its inception, reliability engineering has employed statistical and probabilistic models to assess the reliability and integrity of components and systems. To supplement the traditional techniques, an empirically-based approach, called physics of failure (PoF), has recently become popular. The prerequisite for a PoF analysis is an understanding of the mechanics of the failure process. Entropy, the measure of disorder and uncertainty, introduced from the second law of thermodynamics, has emerged as a fundamental and promising metric to characterize all mechanistic degradation phenomena and their interactions. Entropy has already been used as a fundamental and scale-independent metric to predict damage and failure. In this paper, three entropic-based metrics are examined and demonstrated for application to fatigue damage. We collected experimental data on energy dissipations associated with fatigue damage, in the forms of mechanical, thermal, and acoustic emission (AE) energies, and estimated and correlated the corresponding entropy generations with the observed fatigue damages in metallic materials. Three entropic theorems—thermodynamics, information, and statistical mechanics—support approaches used to estimate the entropic-based fatigue damage. Classical thermodynamic entropy provided a reasonably constant level of entropic endurance to fatigue failure. Jeffreys divergence in statistical mechanics and AE information entropy also correlated well with fatigue damage. Finally, an extension of the relationship between thermodynamic entropy and Jeffreys divergence from molecular-scale to macro-scale applications in fatigue failure resulted in an empirically-based pseudo-Boltzmann constant equivalent to the Boltzmann constant.

## 1. Introduction

Prognostics and health management (PHM) is a promising method in reliability engineering to supplement traditional life assessments. The traditional damage measurements in fatigue, for example, crack growth and load-carrying capacity reduction, are detectable only in the later stages of life and are ineffective in characterizing damage during the earlier periods of life [1]. In contrast, PHM-based life estimation and prognosis incorporates related monitored damage variables into deterministic physics of failure (PoF) models [2,3,4,5,6]. In data-driven prognostics in PHM, observed damage precursors, such as initiation of very small cracks, are collected during system operation and are used to estimate the so-called remaining useful life (RUL) [5,7]. The approaches used to meet the requirements of early life prediction include the uses of entropy. Examples of entropic theories of damage for life prediction include the degradation-entropy generation (DEG) theorem [8] and the principle of maximum entropy (PME) [9,10,11,12,13]. The maximum entropy (MaxEnt) distribution, according to the PME, is the best choice to capture the state of knowledge and information about damage (e.g., measured PHM data). However, thermodynamic entropy, according to the DEG theorem, offers a direct representation of damage [8,14]. Both entropic representations offer powerful foundations for early life fatigue prediction.

Entropy, according to the DEG theorem, is based on irreversible thermodynamics and can be used to depict the endurance to failure, such as cycles to crack initiation or fracture [14,15]. Pioneering works in entropic approaches have verified successful applications to several failure mechanisms, such as fatigue, corrosion, and wear. These entropies are derived from sources of irreversible energy dissipation [14,15,16,17]. In the case of fatigue damage, irreversible energy dissipations include plastic mechanical work, heat, and acoustic emission [18]. A popular entropic approach in fatigue is to use plastic strain energy and surface temperature [19,20]. In this approach, the existence of a fixed entropic endurance, irrespective of the underlying conditions that lead to fatigue damage and failure, is experimentally verified. It has resulted in good agreement with the DEG theorem. Another approach has used acoustic energy dissipation during fatigue in the form of generated acoustic emission (AE) waveforms, where associated information entropy typically correlates well with the amount of the fatigue damage [21]. 

Strain energy dissipation during the cyclic fatigue loading and unloading also appears to apply to the relative entropy. Crooks et al. [22] have shown that the Kullback–Leibler divergence computed from loading/unloading distributions is equivalent to the thermodynamic entropy when distributions of loading/unloading processes are measurable. This concept was demonstrated by Collin et al. [23], who measured thermal dissipation in the unfolding/folding process of a ribonucleic acid (RNA) strand. Loading/unloading work distributions were also used by Douarche et al. [24] to measure a brass wire’s cyclic torsional work and assess the Helmholtz free energy difference. In practical applications, relative entropy in cyclic mechanical work can be computed without the need for temperature information, which provides a potentially simpler entropic damage assessment than the classical thermodynamics.

This paper presents the entropic damage measurements from dogbone coupons that were fatigue tested using three energy dissipations: Plastic mechanical work, thermal energy, and AE. In these approaches, uses of the classical thermodynamics, information (Shannon), and relative entropy were evaluated and discussed in the context of PHM applications. In the proposed approach, relative entropy uses in fatigue damage is new; however, the paper also compares the relationship between these three entropic measures and discusses their applicability to fatigue failures. 

In the remainder of this paper, Section 2 provides reviews of the three entropic theorems and discusses the relative entropy in the context of the fatigue damage process. Section 3 presents the experimental setup, including specimen design, cyclic loading conditions, sensor attachments, and data collection. Section 4 presents the results that support the DEG theorem demonstrated by each entropic approach and discusses the applications and results. Finally, the conclusion section summarizes the results. 

## 2. Fatigue Damage Evaluation Using Three Entropy Measures

According to Lemaitre [1], measurements of fatigue damage include changes detected in crack length, elastic modulus, micro-hardness, ultrasonic wave, and electric resistance. These measurements, also called the markers of damage, are often only detectable when 10–20% of life remains, which is too late for effective prognostic and corrective actions [7]. Therefore, during the early period of life, the assessment of damage must rely on deterministic life models, which tend to be highly uncertain, variable, and conservative [16]. 

Amiri and Modarres [16] have summarized and delineated fatigue damage scales into nano-, micro-, meso-, and macro-scales. Thus, the damage measurement scale evolves from the very small to larger scales, and it is only in the macro-scale that damage can be detected. As such, the lack of detectable damage is highly scale-dependent. However, the damage measurement through the second law of thermodynamics suggests a universal methodology that applies to all the scales discussed above. 

Entropic metrics of damage have been proposed and utilized in engineering applications. Basaran [17,25] and Bryant [14] introduced thermodynamic concepts to assess damage in specific failure modes. Based on irreversible thermodynamic processes, these studies considered the degradation-induced dissipated energy or entropy as a reflection of the cumulative damage process. Amiri and Modarres [16] reviewed entropy for various failure mechanisms, including fatigue, corrosion, and wear, and discussed the corresponding irreversible thermodynamic forces and fluxes used to calculate entropy generation. They reviewed in more detail the DEG theorem, including the concept of entropic endurance introduced by Imanian and Modarres [26]. Experimental results have supported this theorem for fatigue failures [15,19,20] by demonstrating that fatigue fracture occurs at a relatively fixed entropic endurance level regardless of the underlying loading profiles. Figure 1 presents an example confirming this entropic theorem. 

From the irreversible thermodynamics, the dissipative entropy generation may be expressed in the form of Equation (1) [15,16,26]:(1)$$\sigma =\underset{i}{\sum }X_{i}J_{i}$$

This equation is bilinear, where $X_{i}$ is the thermodynamic force and $J_{i\;}$ is the flux due to the dissipation mechanism *i*. Depending on the sources of energy dissipation, Amiri and Modarres [16] presented the entropy generation in its most general form, as shown in Equation (2) [15]: (2)$$\sigma =\frac{1}{T^{2}}\;J_{q}\cdot \nabla T-\sum_{k}J_{k}\left(\nabla \frac{\mu_{k}}{T}\right)+\frac{1}{T}\tau :\dot{\varepsilon_{p}}+\frac{1}{T}\sum_{j}\nu_{j}A_{j}+\frac{1}{T}\sum_{m}c_{m}J_{m}\left(-\nabla \psi \right)$$
where σ is the entropy generation rate, $J_{q}$ is the thermodynamic flux due to heat conduction, $J_{k}$ is the thermodynamic flux due to diffusion, $\mu_{k}$ is the chemical potential, τ is the mechanical stress, $\varepsilon_{p}$ is the plastic strain, $\nu_{j}$ is the chemical reaction rate, $A_{j}$ is the chemical affinity, $c_{m}$ is the coupling constant, $J_{m}$ is the thermodynamic flux due to the external field, and ψ is the potential of the external field.

In this equation, the five terms on the right-side are sources of thermodynamic entropy generation that include heat, diffusion, mechanical work, chemical reaction, and external field effect, respectively. In the fatigue damaging process, heat and mechanical work terms are involved. Naderi et al. [19] numerically calculated the dissipative entropy by using only the mechanical work term, assuming that plastic deformation is the dominant term and the heat conduction effect is negligible, as presented in Figure 1. This assumption was also empirically verified by Imanian et al. [15] and Ontiveros et al. [20], In addition, the concept of entropic endurance was further confirmed by Imanian et al. [15], who measured the interacting thermodynamic forces in a coupled failure mechanism, corrosion-fatigue. Summation of entropies from both mechanical work (fatigue) and chemical reactions (corrosion) contributed to the total entropic endurance at the point of fatigue failure.

In addition to the heat and mechanical work, AE has been considered another source of irreversible energy dissipation. Kahirdeh and Khonsari [18] regarded AE absolute energy as an AE waveform feature and a damage indicator. However, the entropic approach was not investigated as a part of their AE-based damage research. The recorded data from the AE sensor was digitized into the so-called waveforms. The AE information (Shannon) entropy may be characterized by the associated probability distribution in the form of a histogram representing each recorded waveform. Hughes [27] introduced information entropy from digitized waveform data collected from ultrasonic tests. Likewise, for specific features of the waveforms, such as the count rate, information entropy has been applied to AE waveforms to assess the entropy of the waveform signals and empirically establish any correlation between the increasing entropy and the ensuing progression of the fatigue damage observed. Digitized data is processed to a corresponding discrete histogram (expressed in $p\left(x_{i}\right)$), and the entropy is computed using Equation (3):(3)$$S=-\sum_{\;i}p\left(x_{i}\right)\log p\left(x_{i}\right)$$

Sauerbrunn et al. [21] used Equation (3) to calculate information entropy using collected AE waveforms from many fatigue tests. In their research, the AE waveform was shown to be a more appropriate damage indicator than the traditional AE features, such as count and energy. 

In addition to the thermodynamic entropy and information (Shannon) entropy, the third approach to entropic damage explored as a new damage metric in this paper relies on the statistical mechanics definition of entropy, which provides relative entropy from energy dissipation modes during the fatigue damage process. Forward and reverse work distribution functions applied during the cyclic loading in fatigue can be related to the thermodynamic work and free energy. The so-called Crooks fluctuation theorem expressed in Equation (4) is one such relationship [28]: (4)$$\frac{\pi_{f}\left(+W\right)}{\pi_{r}\left(-W\right)}=\exp\left[\frac{W-\Delta F}{k_{B}T}\right]$$
where $\pi_{f}\left(+W\right)$ and $\pi_{r}\left(-W\right)$ in the content of the fatigue damage process may be interpreted as the forward and reverse work distributions over many load cycles, respectively, W is the net strain energy dissipated, ΔF is the Helmholtz free energy difference, $k_{B}$ is the Boltzmann constant ($1.381\times 10^{-23}$ J/K), and T is the temperature. Equation (4) has been applied to nano-scale systems, such as ribonucleic acid (RNA) strands, by introducing forward/reverse work to measure the Helmholtz free energy difference (ΔF) as the RNA system’s inherent property [23]. This paper introduces an extension of this notion into a macro-scale system (i.e., fatigue) and examines its consistency with a fatigue damage assessment based on traditional thermodynamic entropy and information entropy. 

By using the second law of thermodynamics and the Helmholtz free energy definition, Equation (4) can be converted to calculate the total entropy, as shown in Equation (5): (5)$$\Delta S_{tot}=k_{B}\ln\left(\frac{\pi_{f}\left(+W\right)}{\pi_{r}\left(-W\right)}\right)$$

According to the fluctuation theorem, the unloaded/fully-loaded points should be determined in thermodynamic equilibrium, whereas the loading/unloading in the fatigue process does not require the equilibrium condition. In addition, the source of the fluctuation is only thermal energy dissipation. However, these conditions may be invalid when applied to the macro-scale fatigue damage evaluation. Both the thermodynamic conditions and the mathematical implementation of Equation (5) may have limitations. Regardless of the unsettled extension of this theorem to the macro-scale, this research is inspired by the forward/reverse work convention and seeks to empirically investigate the application of this notion to assess fatigue damage. Crooks and Sivak [22] discuss measures of the trajectory ensemble. Consistent with Crooks and Sivak results, relative entropy and Jeffreys divergence (JD) effectively capture the symmetric hysteresis properties of the fatigue phenomenon. Furthermore, in the molecular-scale, JD is related to the classical thermodynamic entropy through the Boltzmann constant.

The relative (divergence) entropy in continuous distribution form is shown in Equation (6) [22]: (6)$$D(\pi_{f}\;|\;\pi_{r})=\int \pi_{f}\left(+W\right)\ln\left(\frac{\pi_{f}\left(+W\right)}{\pi_{r}\left(-W\right)}\right)dW$$

Relative entropy may be interpreted in the classical thermodynamics as the total entropy difference [22]: (7)$$D(\pi_{f}\;|\;\pi_{r})=\frac{1}{k_{B}T}\left(\langle W_{diss}\rangle {}_{f}\right)=\frac{1}{k_{B}T}\left({\langle W\rangle }_{f}-\Delta F\right)=\frac{1}{k_{B}T}\left({\langle W\rangle }_{f}-\Delta {\langle E\rangle }_{f}+k_{B}T\Delta S_{f}^{sys}\right)\;=-\frac{1}{k_{B}T}{\langle Q\rangle }_{f}+\Delta S_{f}^{sys}=\Delta S_{f}^{env}+\Delta S_{f}^{sys}=\Delta S_{f}^{tot}$$
where, in the nano-scale, $k_{B}$ is the Boltzmann constant ($1.381\times 10^{-23}$ J/K), T is the temperature, ${\langle W\rangle }_{f}$ is the mean work in the process *f* (forward work), $\langle W_{diss}\rangle {}_{f}$ is the mean dissipative work, $\Delta {\langle E\rangle }_{f}$ is the mean internal energy difference, ${\langle Q\rangle }_{f}$ is the mean heat dissipation, $\Delta S_{f}^{sys}$ is the entropy change within the system, $\Delta S_{f}^{env}$ is the entropy dissipated to the environment, and $\Delta S_{f}^{tot}$ is the total entropy during the process *f*. The relative entropy in the process *f* is interpreted as the product of the thermodynamic dissipative work $\left({\langle W\rangle }_{f}-\Delta F\right)$ and the constant $\left(\frac{1}{k_{B}T}\right)$. Consistent with its definition, the Helmholtz free energy difference, ΔF, expands to the sum of internal energy ($\Delta {\langle E\rangle }_{f}$) and the product of system entropy difference ($\Delta S_{f}^{sys}$) and the constant ($-k_{B}T$). Considering the first law of thermodynamics, the mean work and mean internal energy difference become the product of the mean heat dissipation (${\langle Q\rangle }_{f}$) and the constant $\left(-\frac{1}{k_{B}T}\right)$, which is expressed in terms of the entropy difference dissipated to the environment. Therefore, the relative entropy in the process *f* is expressed by the total entropy difference $\left(\Delta S_{f}^{tot}\right)$. The relative entropy of the reverse process is: (8)$$D(\pi_{r}\;|\;\pi_{f})=\frac{1}{k_{B}T}\left(\langle W_{diss}\rangle {}_{r}\right)=\frac{1}{k_{B}T}\left({\langle W\rangle }_{r}+\Delta F\right)=\frac{1}{k_{B}T}\left({\langle W\rangle }_{r}-\Delta {\langle E\rangle }_{r}+k_{B}T\Delta S_{r}^{sys}\right)\;=-\frac{1}{k_{B}T}{\langle Q\rangle }_{r}+\Delta S_{r}^{sys}=\Delta S_{r}^{env}+\Delta S_{r}^{sys}=\Delta S_{r}^{tot}$$

For the reverse process r, it should be noted that, unlike the forward process, the Helmholtz free energy difference, ΔF, should be expressed with the positive sign.

Summing Equations (7) and (8) is defined as the JD and represents the dissipative thermodynamic entropy as related to the hysteresis associated with the cyclic loadings in fatigue [22]: (9)$$\begin{array}{ll} Jeffreys\left(\pi_{f};\pi_{r}\right) & =D\left(\pi_{f}\;|\;\pi_{r}\right)+D\left(\pi_{r}\;|\;\pi_{f}\right)=\Delta S_{f}^{env}+\Delta S_{f}^{sys}+\Delta S_{r}^{env}+\Delta S_{r}^{sys}\\ & =\Delta S_{f}^{env}+\Delta S_{r}^{env}=\Delta S^{env}. \end{array}$$

In Equation (9), the terms $\Delta S_{f}^{sys}$ and $\Delta S_{r}^{sys}$ are canceled out, and the only term remaining is the dissipative entropy. Therefore, JD, from the statistical mechanics, corresponds to the thermodynamic entropy as described in the classical thermodynamics. Additionally, JD is only computed by strain energy distributions in fatigue.

## 3. Experimental Setup and Fatigue Damage Entropy Analyses

### 3.1. Specimen Preparation: Design, Evaluation, Manufacturing, and Surface Processing

In a series of uniaxial tensile fatigue experiments, stainless steel (SS) 304L was selected as the testing material. SS304L is a widely used structural material, especially in highly acidic environments. The properties of this alloy are shown in Table 1. The dogbone-shape specimen was selected and designed for fatigue testing under the American Society for Testing and Materials (ASTM) 406 guidelines [29]. To induce the crack formation at the center of the specimen, a V-shaped notch with $K_{T}=4.04$ was designed. The stress concentration factor was calculated using a Peterson’s plot, provided on the efatigue.com website [30]. The V-shape notch, which has a higher concentration-effect than a round-shaped notch, was selected in order not to have the crack around the loading hole. The V-shape notch was designed to minimize the AE noise by reducing the contact area from the noise source. After the design was selected, uniaxial stress distribution was investigated using the finite element method (FEM) with the ANSYS Workbench version R16.2 [31]. The maximum stress was detected at the notch center as expected, and no abnormal stress was found throughout the specimen geometry. Figure 2 shows the shape and dimensions of the specimen. 

The specimens were manufactured using electrical discharge machining (EDM). A total of 50 specimens were prepared for the series of tests, i.e., five loading conditions and 10 test repetitions. After cutting out the specimens, the specimen surface around the crack growth area was processed to clarify the surface image. First, the surface was sanded with increasingly larger grit numbers (grit # 400 → 800 → 2000), then the surface was polished with a polishing pad using one μm alumina solution. Finally, the etching process was employed using a Carpenters etchant.

### 3.2. Cyclic Loading Process

In this uniaxial loading test, a servo-hydraulic testing system was used. An Instron 8800 system was retrofitted on an MTS 311.11 frame. Each specimen was held and loaded by upper and lower wedge grips, and the actuator was connected to the lower wedge grip to apply cyclic tensile loading. The loading conditions were in the range of 16–24 kN maximum loads, 0.1 stress (or loading) ratio, and 5 Hz frequency. After every 1000 cycles, the cyclic loading was paused and clear microscope images were taken. Each test stopped at the pre-set limit of the actuator position (+1.5 mm). Table 2 summarizes the loading conditions.

### 3.3. Measurement Setup

#### 3.3.1. Stress and Strain

Load and extension data were collected by the Instron 8800 system [32,33]. A LEBOW 3116-103 load cell monitored loading applied in the specimen, and an Epsilon extensometer model 3542 measured the extension. The gauge length was 25 mm, and several rubber bands attached the extensometer to the specimen, centering it over the specimen’s notch. The Instron 8800 system tabulated the load and extension data with 200 Hz frequency. The raw data of the load and extension were converted to stress and strain using the specimen geometry information (e.g., the cross-sectional area).

#### 3.3.2. Acoustic Emission

Two Physical Acoustics Micro-30 s resonant sensors were symmetrically attached to the specimen surface 23 mm from the specimen center. The symmetric sensor placement made it possible to apply the delta T filtering technique [34] to filter out the AE signals generated, other than the area of interest. The electric signal from the piezoelectric AE sensors was amplified by the preamplifier in 40 dB gain mode. Overall control and recording of the AE signal were operated by AEWin SW [34].

#### 3.3.3. Surface Temperature

A thermocouple (Omega 5TC-TT-K-40-36) [35] was attached to the surface of the specimen (close to the notch tip). The thermocouple was connected to a National Instrument 9211A module and controlled by NI Labview software [36]. The surface temperature was recorded every half second.

#### 3.3.4. Crack Length Measurement

During the fatigue tests, an optical microscope system (Edmond 2.5–10X microscope body combined with OptixCam Pinnacle Series CCD digital camera) took images of the crack growth area. Images were taken every 5 s, controlled by OCView SW [37]. Every 1000 cycles, crack initiation and propagation were investigated. The crack length was monitored to collect data on the observable damage, and the material fatigue life was defined as specific crack lengths, e.g., 250 μm. 

### 3.4. Data Analysis: Calculating Entropies

After the tests, entropies were calculated using the collected data. Acoustic emission waveform data were sorted after filtering, and the valid waveforms were converted to information entropy according to the equations discussed in Section 2. From the load and extension data, plastic strain energy was computed for each cycle. Classical thermodynamic entropy was calculated by combining surface temperature with the corresponding plastic strain energy. Jeffreys divergence was computed by relying on forward/reverse work distributions, of which the data were collected within the same test groups and the same proportions of life. 

## 4. Results and Discussion

In this section, classical thermodynamic entropy is verified with its entropic endurance, then JD from the forward/reverse work distribution is computed, evaluated, and compared to the classical thermodynamic entropy results. Information (Shannon) entropy of the detected AE waveform is also computed, and its correlation with the classical thermodynamic entropy results is discussed.

### 4.1. Classical Thermodynamic Entropy (CTE)

#### 4.1.1. Entropy Calculation Process

As described in Equation (2), thermodynamic entropy generation is computed by the bilinear equation of force and flux for each energy dissipation mode. In the fatigue damage process, mechanical work is the dominating term, as experimentally proved from previous studies [14,20,26]. Plastic strain energy is computed numerically using discrete stress-strain data. Figure 3 illustrates the process of plastic strain energy calculation for each cyclic loading. Summation of the forward and reverse work (strain energy) makes up the plastic strain energy. This forward/reverse work convention is further used in the JD calculation.

Temperature, measured by the thermocouple, was recorded every half second during each test. As an example, Figure 4 shows the temperature measurement of the test 8VA03. After acquiring both strain energy and temperature, classical thermodynamic entropy was calculated based on the third term of Equation (3) for each cycle.

#### 4.1.2. Results and Evaluation of Classical Thermodynamic Entropy

Figure 5a presents the cumulative classical thermodynamic entropy for a series of 10 tests with 22 kN maximum loading (i.e., tests 8VA13–22). For each cumulative entropy plot, the initial trend is nearly linear, then the slope rapidly increases. Using the calculated life data determined by the crack length, the cumulative entropy for each life was identified, as shown in Figure 5b. 

Figure 6 presents the cumulative entropy at each defined life by the crack length, with respect to the fatigue loading conditions. Stress amplitude, according to the Smith–Watson–Topper (SWT) equation, was used as the representative fatigue loading condition [38,39]. The effect of the stress amplitude (slope in the regression line) diminishes as the crack length of the defining failure decreases.

The result indicates that entropic endurance has a small positive statistical correlation with the stress amplitude. The extensometer with 25 mm gauge length measured the strain (global strain), and the stress field is assumed to be proportional within the gauging area. This assumption is closer to reality before crack initiation. As the crack grows, the plastic zone area increases, and the stress distribution is more biased toward the plastic zone [38]. Nevertheless, endurances determined from the crack length criteria are also valid in the similar measurement setup applications. A similar entropic endurance behavior was also reported by Ontiveros et al. [20,40,41], who found that the cumulative strain energy or thermodynamic entropy at the crack initiation mildly increases with the stress amplitude.

### 4.2. Jeffreys Divergence: The Entropy of Strain Energy Distributions

#### 4.2.1. Analysis and Results: Distribution of Forward/Reverse Work and JD Calculation

The first step to calculate JD using strain energy is to develop the forward and reverse work distributions. Forward/reverse work data within the same loading condition test group of fatigue tests, and strain energies with the same life ratio, were gathered. In this process, the life (cycles) was determined as a function of crack length, as described in Section 3.3.4. Ten strain energy data (i.e., from each test group of the same loading condition and the same life ratio) were fitted to the 3-parameter MaxEnt distribution [12,13]. Figure 7 shows an example of the estimated forward/reverse work mean and standard deviation with respect to the life ratio, and Figure 8 presents an example of forward/reverse work distributions at a given life ratio based on the estimated MaxEnt distribution parameters [12].

After the parametric estimation for each strain energy data set, relative entropies (both $D(\pi_{f}\;|\;\pi_{r})$ and $D(\pi_{r}\;|\;\pi_{f})$) were computed using Equation (6). The cumulative JD was calculated and plotted, as shown in Figure 9, which presents the cumulative JD for the test group of 16 kN maximum load. Similar to the classical thermodynamic entropy, JD is initially linear, then the slope increases as the crack grows.

#### 4.2.2. Evaluation: Correlation to the Classical Thermodynamic Entropy

In the evaluation of possible fatigue damage measurements, the damage is normalized according to Equation (10) [15,21]: (10)$$D=\frac{M_{i}-M_{0}}{M_{f}-M_{0}}$$
where $M_{0}$ is the measured damage at time 0 or the pristine state of the specimen, $M_{f}$ is the damage at the failure (e.g., fracture), and $M_{i}$ is the damage at a given instance (loading cycle). Depending on which crack length is used to determine the failure, $M_{f}$ was differently determined, meaning that, for example in the case of crack initiation, $M_{f}$ corresponds to the measured damage at that point. The initial application of this damage measure was inspired by the Palmgren–Miner rule [42,43], in which the fatigue damage is measured in the proportion of the number of cycles. Not only the number of cycles, but also several measures, such as crack length, load-carrying capacity, and elastic modulus degradation have been utilized as measures of damage in the normalized damage [1]. Normalized entropic damage was first introduced by Imanian and Modarres [15] and used by Sauerbrunn et al. [21].

Figure 10 shows one of the five test groups (10 tests of 16 kN maximum loading) where normalized cumulative JD is linearly correlated to the normalized reference damage (classical thermodynamic entropy). The correlation between the JD and the classical thermodynamic entropy is consistent except at the point of fracture. All the loading groups present this inconsistency at the fracture failure. In case of large crack lengths, it is shown that the JD underestimates fatigue damage compared to the classical thermodynamic entropy. The cause of this inconsistency needs to be further investigated. 

Jeffreys divergence and thermodynamic entropy in molecular-scale are related through the Boltzmann constant ($k_{B})$. However, in the context of the macro-scale application in fatigue using Equations (5), (7), and (8), classical thermodynamic entropy (CTE) is empirically shown to be related to JD by the means of the pseudo-Boltzmann constant,$\;k_{pB}$, where, in Equation (11), $k_{B}$ changes to $k_{pB}.$

(11)$$CTE=k_{pB}\cdot \;JD$$

The pseudo-Boltzmann constant $k_{pB}$, which no longer has the same interpretation and unit as the Boltzmann constant in our macro-scale application, was computed from the slope of the fitted line relating the cumulative JD to the mean classical thermodynamic entropy, as shown in Figure 11, with the slope summarized in Figure 12.

The application of the fluctuation theorem to the macro-scale energy dissipation in the fatigue test has scale limitations. The comparison of the macro-scale applications in the fatigue tests to the reported RNA test is detailed in Table 3. In our experiments, the fluctuation source was extended from the molecular-scale to the macro-scale by changing the measurement mode from thermal to plastic strain energy in the macro-scale application. In this extension, the fluctuation was assumed to be caused by multi-scale dimensional variability. In our experimental investigations, the fluctuation was presented by the formation of forward/reverse strain energy distributions. Furthermore, the converting factor (namely, the pseudo-Boltzmann constant) shows statistical consistency that further supports our assumption that JD can be empirically applied as an alternative damage measurement. Further empirical surveys need to consider other conditions, such as material, geometry, damage mode, and stress conditions. The pseudo-Boltzmann constant, $k_{pB},$ can be generalized empirically.

### 4.3. AE Information Entropy

AE sensors, attached on the specimen surface, collected acoustic energy dissipation in the form of elastic AE signals (waveform) represented by digitized voltage data. Each waveform file is transformed into its equivalent discrete probability distribution, represented by a histogram, and used to quantify the information entropy, as expressed by Equation (3).

#### 4.3.1. Analysis of Information Entropy (IE)

To calculate information (Shannon) entropy from AE waveform data, we followed the approach reported by Sauerbrunn et al. [21] and Kahirdeh et al. [44], where information entropy is calculated from the discrete histogram of waveforms. Figure 13 presents the procedure for AE information entropy calculation. Variations in the bin size parameter of the histograms of the AE waveforms showed that the maximum entropy would be achieved by the selected bin size.

Figure 14 presents an example of the individual and cumulative information entropies. On the cumulative entropy plot, the crack-length points were marked. It is observed that the cumulative entropy trend becomes far steeper around the point of crack initiation. This change in trend is useful information for PHM applications.

#### 4.3.2. Evaluation of AE Entropy and Correlation with Fatigue Damage

The AE count, absolute energy, and information entropy are compared to the classical thermodynamic entropy, as shown in Figure 15, where the failure is defined at the crack initiation. The overall at-a-glance observation shows that the AE information entropy is the closest to the CTE damage. The mean deviation (mean absolute distance from CTE damage to an AE feature) was computed for each test. The sign test was used to assess AE entropy performance using the mean deviation.

The sign test is a nonparametric statistical test that measure consistent differences between pairs of observations and calculates the tests statistic from the difference in the median of the two populations [45]. In this sign test, the left tail mode was utilized, and the entailed hypotheses are shown in Equation (12) (the sign test expressed in signtest(a,b)): (12)$$H_{0}:\;a-b\geq 0;{\;H}_{1}:a-b<0$$

When the *p*-value from this statistic is less than a significance level (10% in this test), the null hypothesis is rejected, and the result concludes that the median of a is less than that of b.

Table 4 presents the sign test results for all the cases (failure defined by the crack length and AE sensor channel). From the results, one can conclude that the information entropy is better than the count and absolute energy, except for the case of fracture failure, and this result is also consistent with Sauerbrunn et al.’s [21] conclusions. 

### 4.4. Summary and Comparison

In Section 4.1, Section 4.2 and Section 4.3, three entropic approaches were reported for applications to fatigue damage assessment. Classical thermodynamic entropy was assessed in terms of the DEG theorem by presenting the existence of an entropic endurance indicating fatigue failure. The assessments of Jeffreys divergence and AE information entropy were followed using the CTE as the reference damage. From the assessment results, JD and AE information entropy exhibit reasonable correlations to the fatigue damage. Furthermore, JD quantitatively correlates with CTE through the pseudo-Boltzmann constant ($k_{pB}$). Correlation analyses show that JD has a better correlation to the reference damage than the AE information entropy. The analyzed entropic approaches are compared and summarized in Table 5. It is noted that the simulation of the entropic prediction model, for example, through a finite element approach, is more applicable to CTE and JD than the AE information entropy. For example, similar to Mozafari et al. [46], fatigue damage simulation modeling using mechanical plastic deformation can be equally applicable to CTE and JD.

## 5. Conclusions

In this paper, three entropic approaches for application to the metallic material fatigue damage process were explored and experimentally demonstrated. Three energy dissipations resulting from mechanistic degradation phenomena—plastic mechanical strain energy, heat (temperature), and acoustic emission—were monitored in multiple uniaxial cyclic fatigue tests. In these entropic approaches, the measured dissipations were quantified in terms of the classical thermodynamic entropy, Jeffreys divergence representing thermodynamic entropy, and information (Shannon) entropy of AE waveforms. Particularly, the application of Jeffreys divergence concept was extended to the macro-scale and applied to the fatigue damage assessment. Classical thermodynamic entropy showed a consistent entropic endurance to fatigue damage. Further, Jeffreys divergence and AE information entropy were adequately correlated to the fatigue damage. The contribution from this research was in limited extensions and applications of the three entropic methods, which resulted in the following findings:
In classical thermodynamics, the entropic endurance showed a slight correlation with the cyclic stress amplitude. This entropy was shown to be an appropriate index of damage. Application of Jeffreys divergence in macro-scale was empirically explored and computed from the forward/reverse work distributions, which showed an excellent correlation to the normalized damage. The quantitative conversion factor (namely the pseudo-Boltzmann constant, $k_{pB}$) also showed consistency between the classical thermodynamic entropic damage and Jeffreys divergence-based entropic damage. Fatigue damage assessment using information (Shannon) entropy of the acoustic emission waveform data, compared well with the classical thermodynamic entropy. Similarly, using statistical tests, it was shown that the AE-based informational entropy of damage was more consistent than the two conventional AE features (i.e., count and absolute energy) used in the fatigue damage assessment.

## Figures and Tables

**Figure 1 entropy-21-00804-f001:**
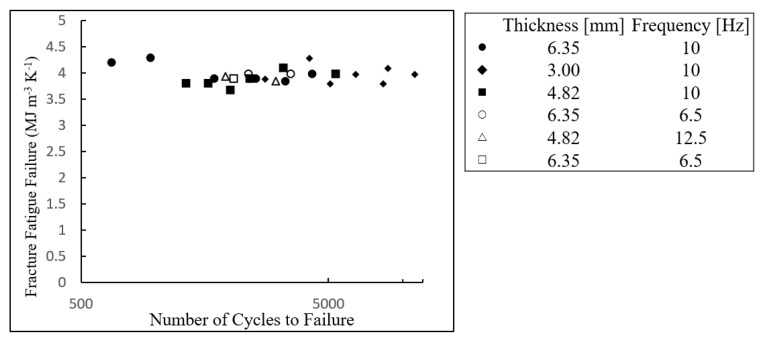
Cumulative entropy for various fatigue test loading conditions. The entropic data points show that the thermodynamic entropy has an endurance level to failure, irrespective of the path to failure. The cyclic bending loading was applied to each specimen with the amplitude of 25–50 mm. Entropic endurance raw data were from Figure 6 of Naderi et al. [19].

**Figure 2 entropy-21-00804-f002:**
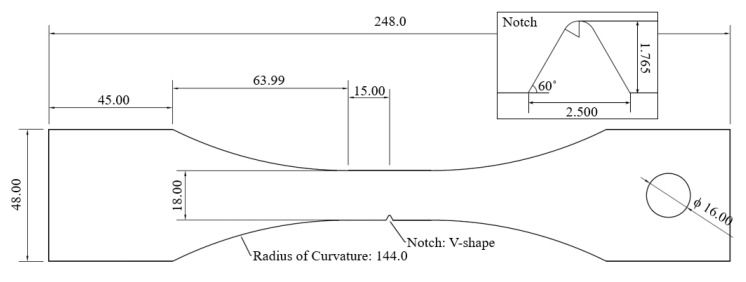
The geometry of the dogbone specimen. The specimen has a hole for loading with a 16 mm diameter pin and stress concentrated by a V-shape notch. Theoretical stress concentration factors ($K_{T}$) are 4.04 for the notch and 3.44 for the hole (pin in tension condition), respectively. The length unit is in millimeters.

**Figure 3 entropy-21-00804-f003:**
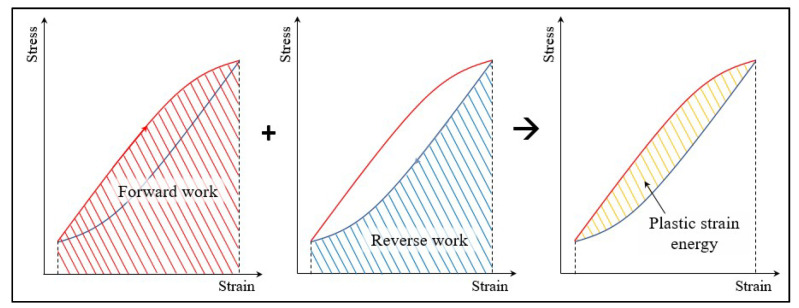
Strain energy calculation procedure. For each cyclic loading, the stress-strain path is divided into forward/reverse work processes, and strain energy is separately computed. The summation of two works is the plastic strain energy or hysteresis.

**Figure 4 entropy-21-00804-f004:**
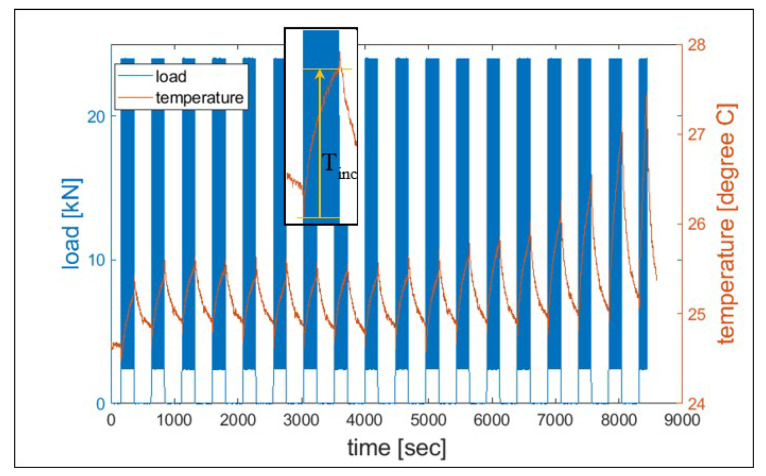
Temperature monitoring during the overall test (8VA03). The ninth damaging loading process is magnified to highlight the temperature rise during the loading process.

**Figure 5 entropy-21-00804-f005:**
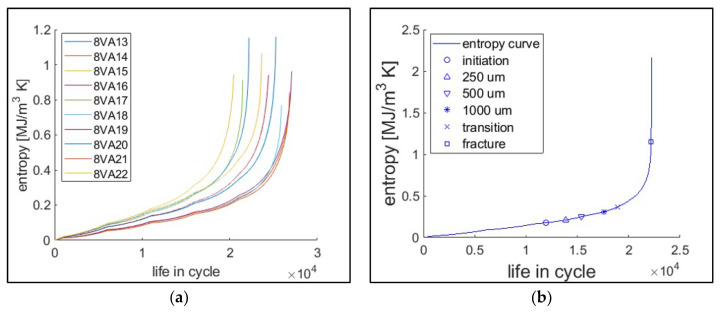
(**a**) Cumulative classical thermodynamic entropy for 10 tests with 22 kN maximum load. (**b**) The cumulative entropy measured by crack growth. After every 1000 cycles, the cyclic loading process was stopped to perform some measurements. This effect is seen as a slight discontinuity in the plotted curves.

**Figure 6 entropy-21-00804-f006:**
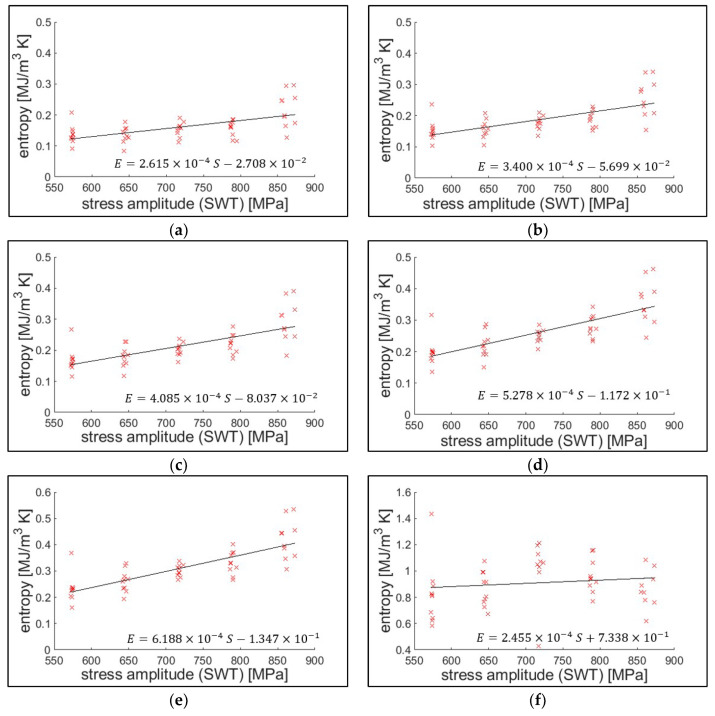
Classical thermodynamic entropy endurance for each defined life under crack growth. The life is determined at (**a**) crack initiation, (**b**) 250 μm crack, (**c**) 500 μm crack, (**d**) 1000 μm crack, (**e**) transition (from region II to III of linear elastic fracture mechanics), and (**f**) fracture, respectively.

**Figure 7 entropy-21-00804-f007:**
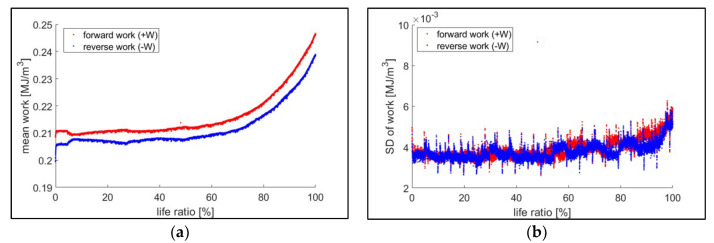
Mean and standard deviation of the collected forward/reverse work data. The data were collected from ten 22 kN maximum loading tests, and the failure (100% life ratio) was determined for an initial fatigue crack length of 1000 μm. (**a**) Shows mean (μ) and (**b**) shows the standard deviation (σ). As noted, standard deviations (SD) of work for forward/reverse normal distributions have a significant overlap.

**Figure 8 entropy-21-00804-f008:**
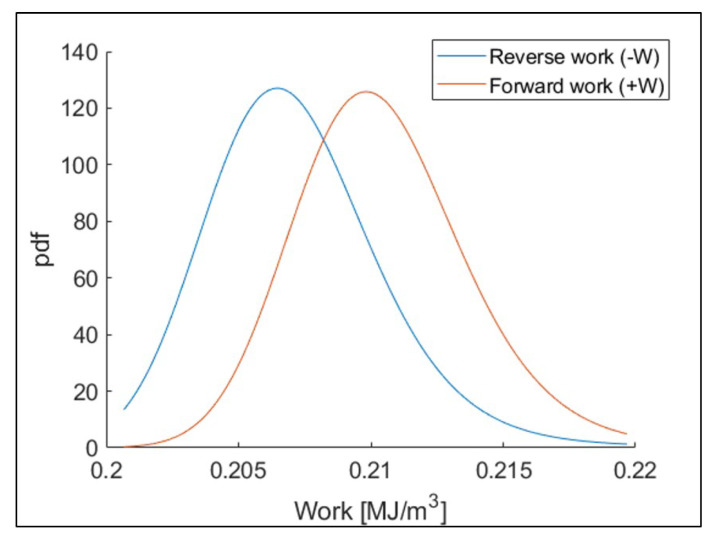
Forward/reverse work distributions of 22 kN maximum loading test group at 25% of life. The distributions were fitted in the maximum entropy (MaxEnt) distribution model.

**Figure 9 entropy-21-00804-f009:**
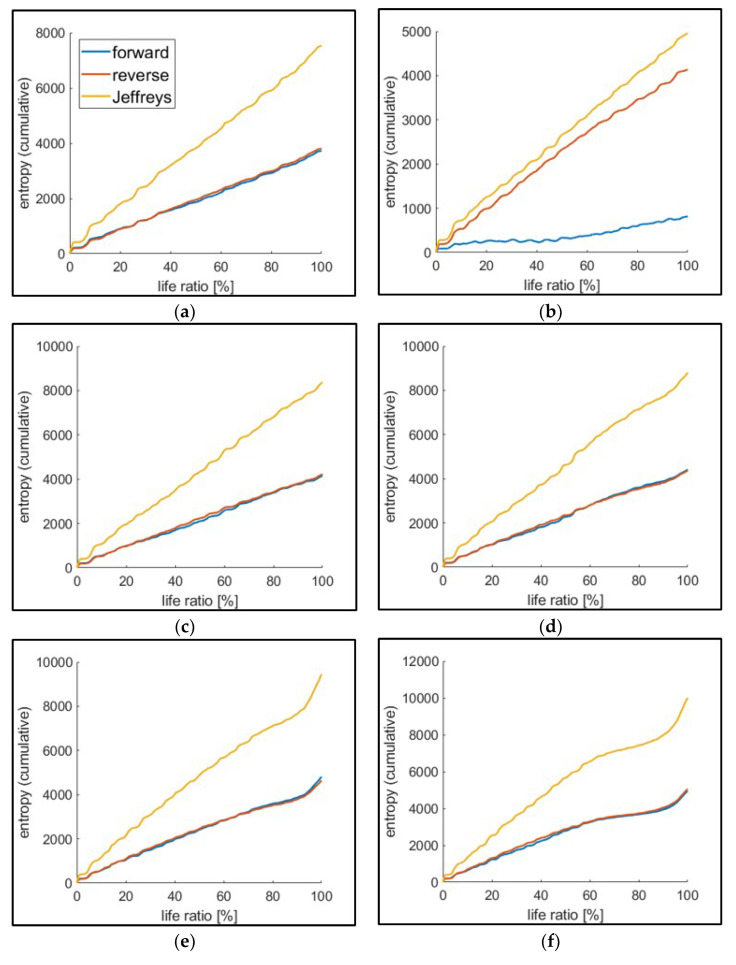
Cumulative relative entropy (Example: For the test group with 16 kN maximum load). Each plot represents the case of normalized life at various crack lengths: (**a**) Crack initiation, (**b**) 250 μm, (**c**) 500 μm, (**d**) 1000 μm, (**e**) transition, and (**f**) fracture.

**Figure 10 entropy-21-00804-f010:**
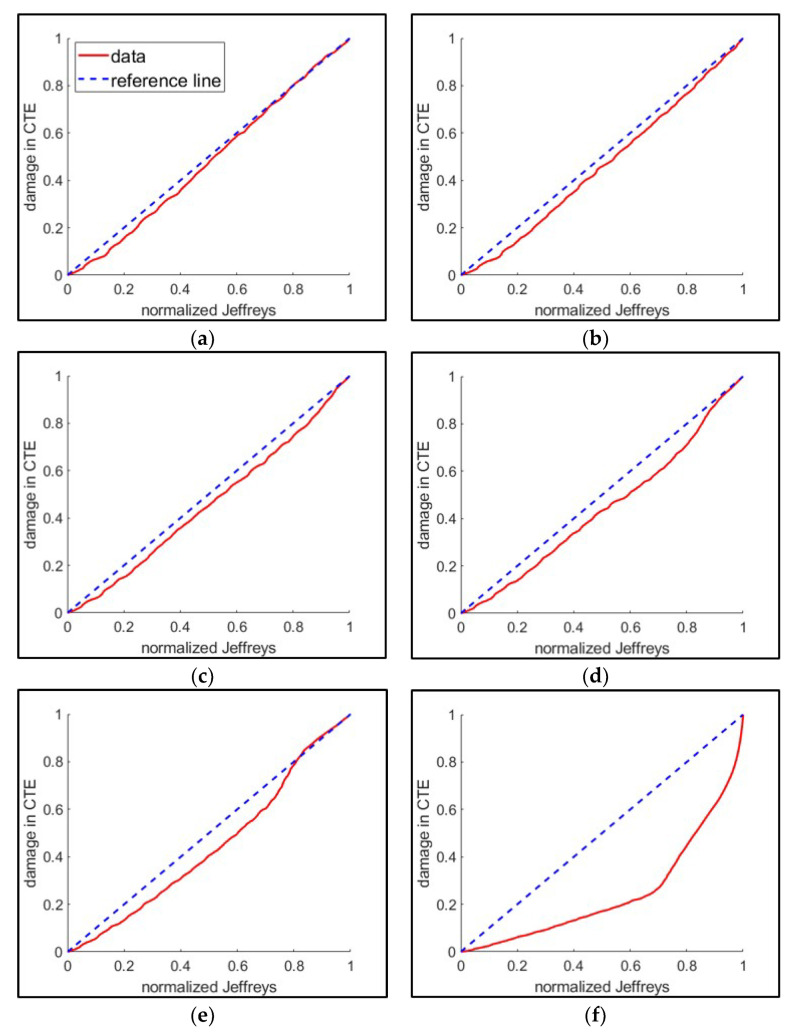
Evaluation of Jeffreys divergence (JD) by correlating to the reference damage (classical thermodynamic entropy (CTE)) as an example of the 16 kN maximum loading test group. Each correlation plot is drawn by the defined point of failure at (**a**) crack initiation, (**b**) 250 μm crack, (**c**) 500 μm, (**d**) 1000 μm, (**e**) transition, and (**f**) fracture.

**Figure 11 entropy-21-00804-f011:**
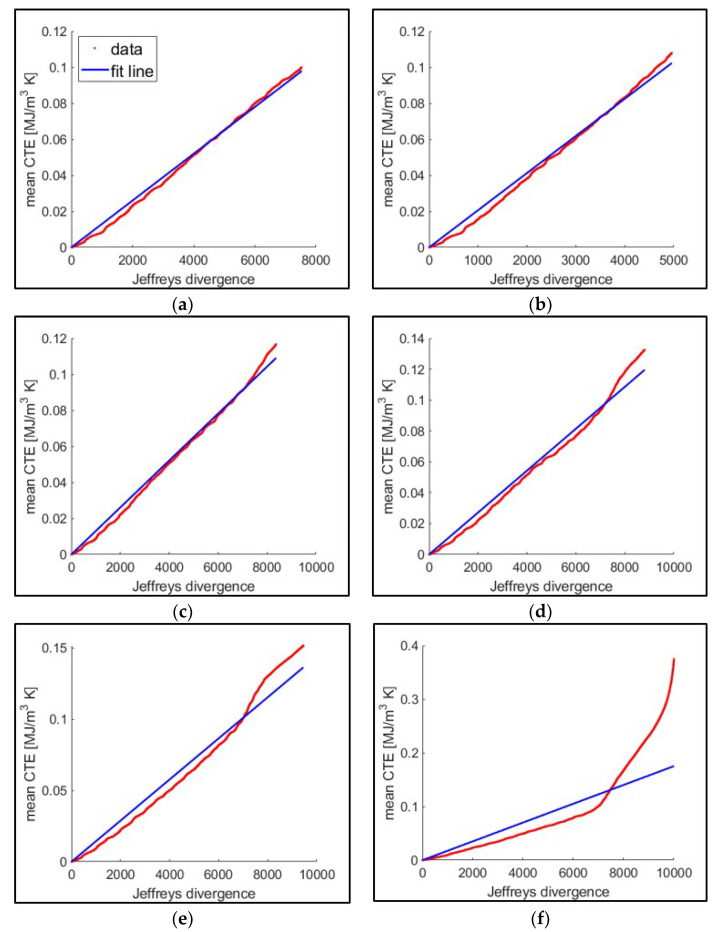
Linear correlation (with the 0 intercept) between mean CTE and JD (for the ten tests of the 16 kN maximum loading group). Using this correlation, the slope is estimated to correspond to $k_{pB}$. Failure is defined at (**a**) crack initiation, (**b**) 250 μm crack, (**c**) 500 μm, (**d**) 1000 μm, (**e**) transition, and (**f**) fracture.

**Figure 12 entropy-21-00804-f012:**
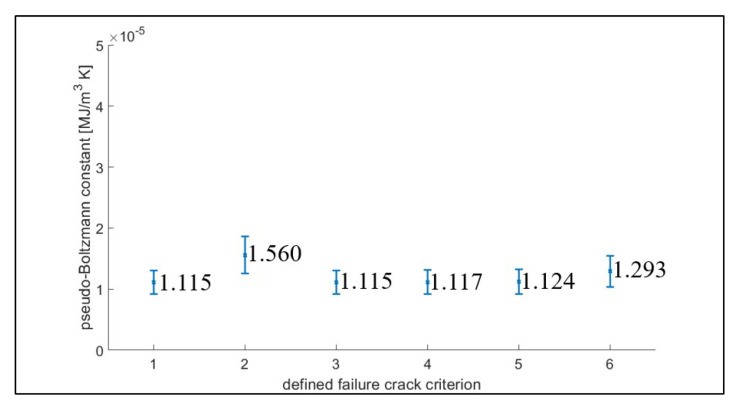
The slope (namely the $k_{pB}$) for each crack-length based failure. The bar of each data point shows one standard deviation above and below the mean shown. Failure is defined as (1) crack initiation, (2) 250 μm crack, (3) 500 μm crack, (4) 1000 μm crack, (5) transition, and (6) fracture.

**Figure 13 entropy-21-00804-f013:**
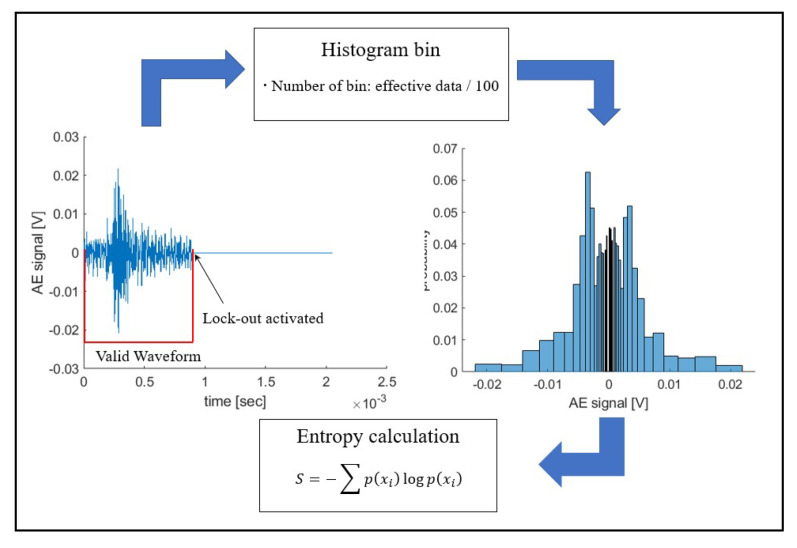
The procedure of AE information entropy calculation. By using the digitized waveform signal data, information entropy is calculated from the generated histogram.

**Figure 14 entropy-21-00804-f014:**
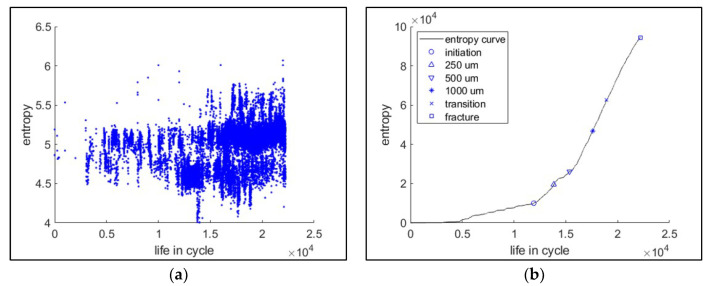
Acoustic emission (AE) information entropy (example: 8VA20). (**a**) Individual entropies for the collected waveforms. (**b**) Cumulative entropy through the life cycle.

**Figure 15 entropy-21-00804-f015:**
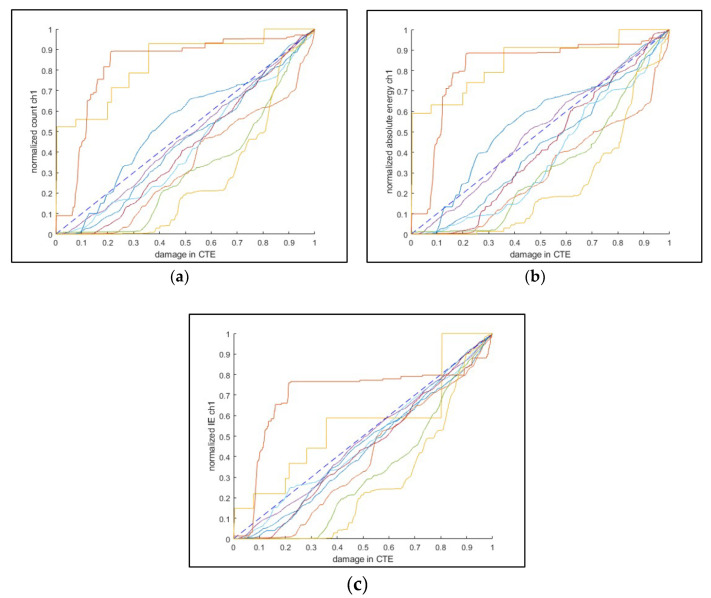
Correlation of AE features to the measured damage (classical thermodynamic entropy). The correlated features are (**a**) count, (**b**) absolute energy, and (**c**) information entropy. These correlation plots were drawn from the 24 kN maximum loading group and AE sensor channel 1 (the sensor more adjacent to the loading actuator).

**Table 1 entropy-21-00804-t001:** Mechanical properties and chemical composition of specimen material (stainless steel (SS) 304L (SS304L)).

Mechanical Properties											
$\sigma_{U}$ [MPa]			$\sigma_{Y}$ [MPa]			Elongation [%]			Hardness [RB *]		
613.8			325.65			54.06			85.00		
**Chemical Composition [w%]**											
C	Cr	Cu		Mn	Mo	N	Ni	P		S	Si
0.0243	18.06	0.3655		1.772	0.2940	0.0713	8.081	0.0300		0.0010	0.1930

* RB: Rockwell hardness measured on the B scale.

**Table 2 entropy-21-00804-t002:** Five test conditions (test group) of uniaxial cyclic loadings. The test groups were categorized based on their maximum loads. Each group consists of 10 specimens tested successfully.

		Max. Load [kN]	Test Specimen IDs
Stress ratio	0.1	16	8VA43–8VA 52
Frequency	5 Hz	18	8VA33–8VA 42
# of cycle per block	1000	20	8VA23–8VA 32
Loading duration	200 s	22	8VA13–8VA 22
		24	8VA03–8VA 12

**Table 3 entropy-21-00804-t003:** Comparison of Crooks fluctuation theorem application to RNA and metal fatigue test.

	RNA [23]	Metal Fatigue Test
Purpose	Finding Helmholtz free energy	Assessing the amount of damage
Source of fluctuation	Thermal energy Fluctuation in atomic distance	Plastic strain energy Multi-scale defects (e.g., point defect, dislocation, volumetric defect, inclusions, grain structure variability)
Test control	Controlled in displacement Thermal equilibrium at both end of displacement points	Controlled tensile load Thermal equilibrium not controlled
Test repetition	Hundreds of times. A specimen was repeated with unfolding/folding process without regarding the damage	10 fatigue tests repeated with a fixed loading condition, and strain energy data grouped in the corresponding damage
Correlating constant (JD to CTE)	Boltzmann constant ($1.381\times 10^{-23}\;J/K$)	Pseudo-Boltzmann constant estimated from tests $1.115-1.560\times 10^{-5}\;J/m^{3}K$ (range of the mean values)

**Table 4 entropy-21-00804-t004:** Sign test results represented in *p*-value. The sign test rejects the null hypothesis (the former is not less than the latter) when the *p*-value is less than the significance level. In the 10% significance level, the cases of not rejecting the null hypothesis are underlined.

Failure Defined at	a:IE b: Absolute Energy		a:IE b: Count	
	ch1	ch2	ch1	ch2
Initiation	$1.0\times 10^{-1}$	$3.3\times 10^{-2}$	$1.0\times 10^{-1}$	$6.6\times 10^{-2}$
250 μm	$7.7\times 10^{-3}$	$7.7\times 10^{-3}$	$6.0\times 10^{-2}$	$6.0\times 10^{-2}$
500 μm	$1.3\times 10^{-3}$	$1.3\times 10^{-3}$	$1.6\times 10^{-2}$	$7.7\times 10^{-3}$
1000 μm	$4.5\times 10^{-5}$	$1.5\times 10^{-4}$	$3.3\times 10^{-2}$	$6.0\times 10^{-2}$
Transition	$1.2\times 10^{-5}$	$1.5\times 10^{-4}$	$3.3\times 10^{-2}$	$6.0\times 10^{-2}$
Fracture	$4.4\times 10^{-1}$	$2.4\times 10^{-1}$	$6.0\times 10^{-2}$	$1.6\times 10^{-1}$

**Table 5 entropy-21-00804-t005:** Comparison of entropic approaches and efficacy as the measure of fatigue damage.

	Classical Thermodynamic Entropy (CTE)	Jeffreys Divergence (JD)	AE Information Entropy (IE)
Analysis of source data	Plastic strain energy Surface temperature	Plastic strain energy	AE waveform
Calculation method	Bilinear irreversible thermodynamic entropy Equation (2)	Fluctuation theorem and relative entropy Equations (7)–(9)	Information theory Equation (3)
Evaluation	Consistent entropic endurance Used as the reference damage	Correlation to normalized measured damage Pseudo-Boltzmann constant ($k_{pB}$)	Correlation to normalized measured damage
Effect	Endurance verified Linear relation to stress amplitude	Endurance verified Consistent $k_{pB}$	Better than AE count and absolute energy. Useful for early life in pre-crack initiation

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
