# Peer review of "Measures of Entropy to Characterize Fatigue Damage in Metallic Materials"

_entropy, 2019, doi:10.3390/e21080804_

Round 1

Reviewer 1 Report

In the manuscript “Measures of Entropy to Characterize Fatigue Damage in Metallic Materials,” the authors study metallic material fatigue process obtained from three associated energy dissipation sources related to fatigue damage in the forms of mechanical, thermal, and acoustic emission (AE) energies. They use Three different Entropy Measures: thermodynamic entropy, Jeffreys divergence representing thermodynamic entropy, and information (Shannon) entropy.

Some minor revisions:

·         line 19, Remove “In this investigation.”

·         Rewrite line 17, because the entropy has already been used to predict damage and failure in several previous works.

·         The results mentioned in the abstract (lines 24 and 25) are very general. Please include the specific findings of the study present.

·         Include a more detail discussion about the Jeffreys divergence (JD).

·         The authors must explain why they only use three of the five entropy measures proposed in reference 13, namely relative entropy (dissipation), Jeffreys divergence (hysteresis), Jensen–Shannon divergence (timeasymmetry), Chernoff divergence (work cumulant generating function), and R´enyi divergence. Since, as it is known, in a thermodynamic context, the maximum work represents the worst-case scenario under a particular protocol, the single realization that creates the largest increase in entropy.

·         Please, include the missing data of reference 4: page 741, ISBN, etc.

Reviewer 2 Report

The authors consider different but measures of entropy generated during damage and their correlation with fatigue damage (both for initiation of cracks and their subsequent cracking). The  entropy values considered are (a) Instantaneous Shannon entropy of acoustic emission signals (AE entropy) and (b) physical entropy or theromodynamic entropy (TE) based on energy dissipated per cycle  and  (c) the statistical mechanics based Jeffreys Divergence( JD)of forward and reverse work.  

The paper is quite interesting and useful for the purpose of using AE sensors for monitoring damage. The very good correlation of JD with thermodynamic entropy (Fig 11) seems to indicate that one or the other can be used. The key point is that it appears that the AE is useful for monitoring crack initiation.

Furthermore, the authors need to comment on the relative efficacy of the methods for specific fatigue monitoring tasks , i.e. how do they compare with each other in identifying crack initiation (it appears that AE information entropy is quite good, but how does it compare to JD or TE? How about subsequent fatigue crack growth? Again, the paper seems to have the information on these questions, but it is not evident from the conclusions whether these items are correct. The authors need to add a short discusson of the comparison between the three (Table 5 compares only the methods but the outcomes i.e. last column, needs to have relative merits/recommendations  for different use cases

As a final note, there is a major difference between the AE signal monitoring and the other two methods. In principle, TE and JD can be applied to the modeling and simulation stage itself (ie they can be incorporated into FEA simulations (see e.g. Mozafari, F., P. Thamburaja, A. Srinivasa, and N. Moslemi. "A rate independent inelasticity model with smooth transition for unifying low-cycle to high-cycle fatigue life prediction." International Journal of Mechanical Sciences (2019).)  whereas AE entropy cannot. The authors need to comment on advisablity of using either TE or JD in simulation, since presumably that would be a significant advantage.

I thus recommend that the paper be published provided the authors address the relatively minor points raised above

Reviewer 3 Report

The present work deals with the problem of measuring damage in metallic materials due to fatigue.  The paper is structured in five sections. The body of the text begins with a clear presentation to the problematic of interest, introducing a pertinent but incomplete bibliographical revision. In the second section the authors present the entropic approaches used to measure damage in a metal species. In this formalism, the sources of entropy originate from the processes of plastic deformation, heat exchange and acoustic emission. The description of the experimental apparatus and the data acquisition process follows in the third section. The fourth section presents the calculation of the three entropy-based damage measures, commenting on their qualitative and quantitative aspects. Final conclusions close the body of the text in the fifth section.

The estimation of damage in metallic specimens is still a relevant and current research problem due to its continuing and increasing importance in different engineering applications. The use of statistical mechanics techniques to deal with macroscope problems, via adequate generalization / adaptation, is a topic of great technical-scientific value, and provides a rational framework for tackling complicated physical problems such as cracking propagation. In addition, the results presented appear promising. In this sense, the present work deserves consideration.

But prior to acceptance of this work, the authors need to clarify some issues. The following points are worthy of comment:

1 - There is a typo in the article title. It is written "Chareacterize" instead of  "Characterise";

2 - It is not clear from the text what the authors' original contribution was in this work. It is extremely important to make that point very clear.

3 - The introduction is fairly clear, but does not make an extensive review of entropy-based fatigue life prognostic methods. In particular, the authors bypass the techniques that are based on the maximum entropy principle, which are gaining more and more space in the literature:

https://www.mdpi.com/1099-4300/18/4/111

https://link.springer.com/article/10.1007%2Fs10845-009-0341-3

https://www.sciencedirect.com/science/article/pii/S0142112319302415

The authors also go over the classical and other state of art prognostic methods presented in the emerging area of Prognostics and Health Management:

https://www.springer.com/gp/book/9789811020315

https://www.springer.com/gp/book/9783662540282

https://www.springer.com/gp/book/9783319558516

https://www.springer.com/gp/book/9783319447407

https://www.wiley.com/en-br/Prognostics+and+Health+Management%3A+A+Practical+Approach+to+Improving+System+Reliability+Using+Condition+Based+Data-p-9781119356653

4 - As the authors themselves admit in the text, employ Crooks fluctuation theorem to describe the ratio between forward and reverse work distributions over many load cycles can be questionable, because the thermodynamic equilibrium is not a necessary condition in fatigue. However, the idea of trying to adapt this result to a macroscopic system is very interesting, which has as evidence the (nearly) constant value of the "Boltzman constant of fatigue". The authors need to elaborate a few more points to reinforce the evidence that this approach is indeed effective. Testing some aspects that they mention in the text (e.g. dependence on material, geometry, etc.) would be a good start. 

Moreover, a further discussion on this new Boltzmann constant would add greatly. This parameter bridges the micro and macro worlds, serving as a scaling factor between microscopic and macroscopic properties. Classically its order of magnitude is 10^{-23} and in the present application the obtained value is approximately 10. This difference of 24 orders of magnitude has implications in the behavior of the system of interest that can be better explored.

5 - Using parametric statistics assuming a normal distribution to estimate forward and backward jobs is a biased procedure, since there is no guarantee that this is the true distribution. A much more conservative and consistent procedure would be to use the data to estimate the low-order moments of the distribution and then employ the maximum entropy principle to infer the distribution based on this available statistical information. Authors can see details about this approach in:

https://www.springer.com/gp/book/9783319543383

https://www.amazon.com/Entropy-Optimization-Principles-Applications-Kapur/dp/0123976707

6 - In the same line as the previous comment, the construction of the acoustical emission histogram is inconsistent since it assumes a normal law with no guarantee from the physics. The MaxEnt formalism can be employed again to improve predictions.

Some minor remaks:

* Section 2
The development of equations 7 and 8 need to be better explained.

* Section 4.1.2 

In the end of the first paragraph is written Figure 6(b), instead of Figure 5(b).

* Resolution of most of the figures must be improved.

Round 2

Reviewer 3 Report

The authors answered most of the questions raised by this reviewer, but on two points the answers provided were not satisfactory. 

1) Regarding the recommendation to include in the introduction other approaches relating entropy and PHM, it is not extremely serious, although it leaves the work incomplete. The authors themselves acknowledge the connection of the themes in the response letter. It is difficult to understand why not make such a diminutive addition to the text.

2) The second point is more serious, it concerns the use of parametric statistics (assuming a distribution) to make inference. As already argued in the first review, the present reviewer has no confidence in the consistency of the statistical results obtained in this way. The authors themselves acknowledge in the response that a non-parametric approach or using MaxEnt would be more consistent, but did not modify the calculations. This point is extremely critical and should be reviewed.

The opinion of the present reviewer is that the two points are important, especially the second, and need to be reviewed before acceptance of the paper.

Round 3

Reviewer 3 Report

The authors satisfactorily answered the questions raised by this reviewer. The recommendation now favors the publication of the article.

I only add one minor comment:

* The labels of the Figures 5a, 7, 9, 10, 11, 12, 13, 14 and 15 are very small, they need to be improved for the final version.